# Singlet-Oxygen-Mediated Regulation of Photosynthesis-Specific Genes: A Role for Reactive Electrophiles in Signal Transduction

**DOI:** 10.3390/ijms25158458

**Published:** 2024-08-02

**Authors:** Tina Pancheri, Theresa Baur, Thomas Roach

**Affiliations:** Department of Botany, Faculty of Biology, University of Innsbruck, 6020 Innsbruck, Austria

**Keywords:** lipid peroxidation, acrolein, 4-hydroxynonenal, Rose Bengal, reactive carbonyl species, non-photochemical quenching, algae, stress metabolism

## Abstract

During photosynthesis, reactive oxygen species (ROS) are formed, including hydrogen peroxide (H_2_O_2_) and singlet oxygen (^1^O_2_), which have putative roles in signalling, but their involvement in photosynthetic acclimation is unclear. Due to extreme reactivity and a short lifetime, ^1^O_2_ signalling occurs via its reaction products, such as oxidised poly-unsaturated fatty acids in thylakoid membranes. The resulting lipid peroxides decay to various aldehydes and reactive electrophile species (RES). Here, we investigated the role of ROS in the signal transduction of high light (HL), focusing on GreenCut2 genes unique to photosynthetic organisms. Using RNA seq. data, the transcriptional responses of *Chlamydomonas reinhardtii* to 2 h HL were compared with responses under low light to exogenous RES (acrolein; 4-hydroxynonenal), β-cyclocitral, a β-carotene oxidation product, as well as Rose Bengal, a ^1^O_2_-producing photosensitiser, and H_2_O_2_. HL induced significant (*p* < 0.05) up- and down-regulation of 108 and 23 GreenCut2 genes, respectively. Of all HL up-regulated genes, over half were also up-regulated by RES, including *RBCS1* (ribulose bisphosphate carboxylase small subunit), NPQ-related *PSBS1* and *LHCSR1*. Furthermore, 96% of the genes down-regulated by HL were also down-regulated by ^1^O_2_ or RES, including *CAO1* (chlorophyllide-*a* oxygnease), *MDH2* (NADP-malate dehydrogenase) and *PGM4* (phosphoglycerate mutase) for glycolysis. In comparison, only 0–4% of HL-affected GreenCut2 genes were similarly affected by H_2_O_2_ or β-cyclocitral. Overall, ^1^O_2_ plays a significant role in signalling during the initial acclimation of *C. reinhardtii* to HL by up-regulating photo-protection and carbon assimilation and down-regulating specific primary metabolic pathways. Our data support that this pathway involves RES.

## 1. Introduction

Solar energy drives photosynthesis, which is fundamental for sustaining almost all life on earth. Photosynthesis starts with the capture of light energy by pigments in the protein complexes of thylakoid membranes and its transfer to chlorophyll in the reaction centres of photosystems (PS) for initiating photochemistry. Photosynthetic activity requires a variety of unique processes not found in non-photosynthetic organisms (heterotrophs). In total, 597 nucleus-encoded proteins have been identified in green lineage organisms, but not, or poorly conserved, in heterotrophs, which have collectively been grouped as GreenCut2 [1]. About 30% of GreenCut2 genes are of prokaryotic origin, with about half having an unknown function and most encoding proteins predicted to be localized to the chloroplast [2]. Rational for inclusion in GreenCut2 was a putative protein orthologue in the green lineage eukaryotes (Viridiplantae), including the algae *Chlamydomonas reinhardtii* and *Ostreococcus taurii*, the moss *Physcomitrella patens*, and the vascular plant *Arabidopsis thaliana*, not found in heterotrophs (e.g., *Pseudomonas aeruginosa*, *Sulfolobus solfataricus*, *Caenorhabditis elegans*, *Homo sapiens*, etc.) [1].

Molecular oxygen, a product of oxygenic photosynthesis, can form unstable intermediates called reactive oxygen species (ROS). It is well established that across the diversity of life, the many signalling networks functioning in development, or sensing physiological state and environmental change, integrate redox components involving ROS [3,4]. Organelles, including mitochondria, chloroplasts and peroxisomes, are major sites of ROS production in plant cells, which can invoke responses not only locally, but also in the nucleus by activating or suppressing transcription. Here, it is important to distinguish the species of ROS, since they can have significantly different chemical properties. For example, hydrogen peroxide (H_2_O_2_) signalling filters through catalytic sites of peroxiredoxins, thioredoxins and glutaredoxins to trigger target thiol/disulfide redox ‘switches’ [5]. In contrast, singlet oxygen (^1^O_2_) is short-lived, and the associated signalling is largely in response to the oxidation of, for example, carotenoids (e.g., β-carotene), proteins (e.g., EXECUTER1) or cellular components (e.g., membranes) [6]. Increases in H_2_O_2_ and ^1^O_2_ production of *C. reinhardtii* have been observed in response to a 5-fold increase in light intensity [7,8], and could be involved in acclimation to high light (HL) by affecting the transcription of GreenCut2 genes. 

In response to HL, photosynthetic organisms activate non-photochemical quenching (NPQ), which can safely dissipate excess light energy to heat [9]. However, NPQ has limits and ROS formation is an unavoidable part of photosynthesis. In particular, ^1^O_2_, formed by charge recombination in PSII, is considered the most damaging ROS produced under excess light [10,11]. It has become apparent that ^1^O_2_ likely has a role in HL signal transduction [6,12,13,14,15,16,17]. One of the most important components of thylakoid membranes that house photosynthetic complexes are galactolipids, of which a significant fraction is composed of the poly-unsaturated fatty acid (PUFA) linolenic acid [18]. The three electron-rich C=C bonds of linolenic acid are prone to non-enzymatic oxidation by ^1^O_2_, thereby forming lipid peroxides [19]. These can spontaneously dissociate, releasing a variety of carbonyls, including short-chain α,β-unsaturated aldehydes called reactive electrophile species (RES), including acrolein and 4-hydroxynonenal (HNE), which may play a role in HL signalling. 

*Chlamydomonas reinhardtii* was the first Chlorophyte (green alga) to have its genome sequenced [20] and has since become a model organism for investigating HL stress acclimation [17]. In the model angiosperm *Arabidopsis thaliana*, β-cyclocitral from the oxidation of β-carotene has emerged as part of the ^1^O_2_ stress response via the zinc finger protein *methylene blue-sensitive 1* (MBS1) protein [21]. However, despite possessing an MBS1 signalling pathway [22], β-cyclocitral has a minor role in ^1^O_2_ signalling in *C. reinhardtii* [23]. In contrast, the exogenous treatment of low light (LL)-treated *C. reinhardtii* with the RES acrolein was able to lead to a remarkably similar differential expression of genes (DEG) compared with the response to the photosensitizer Rose Bengal, which can be used to mimic HL-induced ^1^O_2_ production [15].

Here, using an RNA seq. approach with *C. reinhardtii*, we investigated the involvement of various ROS-mediated signal-transduction pathways in the HL acclimation of photosynthesis by focusing specifically on the GreenCut2 gene cluster. Comparisons were drawn between the influence of HL and cells treated under LL with H_2_O_2_ or Rose Bengal, the latter of which induced a similar level of lipid peroxidation as occurred under the HL treatment [24]. To investigate potential intermediates in ROS signalling, the response of cells under LL to acrolein, HNE and β-cyclocitral were included. 

## 2. Results

### 2.1. Differental Expression of GreenCut2 Genes in Response to HL

In total, 532 GreenCut2 genes (89% of the total) were identified among the RNA seq. data (Appendix A). Of these, 2 h HL of LL-acclimated cells caused a significant (*p* < 0.05) up-regulation of 86 (Table 1) or 108 genes (Table 2), with or without a false discovery rate (FDR), respectively. 

Four HL-up-regulated genes (Cre01.g016750, Cre01.g016600, Cre10.g440450 and Cre05.g243800) encode four potential PSII subunits (*PSBS*, *PSBS1*, *PSB28* and *PSB27/CPLD45*), and many others have roles in PSII assembly (Table 1). Furthermore, six genes were also upregulated towards chlorophyll synthesis (Cre12.g510050, Cre06.g294750, Cre05.g242000, Cre12.g498550, Cre05.g246800 and Cre01.g042800, encoding *CTH1*, *CHLG*, *CHLD*, *CHLM*, *GUN4* and *DVR1*, respectively; Appendix A), while Cre01.g043350 encoding for *CAO1* for the synthesis of chlorophyll *b* was down-regulated (Table 1). A further HL-up-regulated gene, Cre03.g199535, encoding a low-molecular-mass early light-induced protein (Table 1), is also likely involved in chlorophyll biosynthesis [25]. Indeed, ‘porphyrin and chlorophyll metabolism’ was the only KEGG pathway up-regulated by HL that contained more than three genes (Appendix A). No HL-down-regulated KEGG pathway contained more than one gene (Appendix A).

HL was the only treatment assessed that induced more up- than down-regulation of GreenCut2 genes (Table 2). More total GreenCut2 genes were up-regulated by Rose Bengal (*n* = 140) and acrolein (*n* = 138), but an even greater number were also down-regulated (*n* = 180 and *n* = 114, respectively, Table 2). β-cyclocitral and H_2_O_2_ (from [26]) led to an up-regulation of 2 and 32 GreenCut2 genes, respectively, but <4% overlapped with the response to HL (Table 2).

### 2.2. Shared Genes 

When considering significantly affected genes at *p* < 0.05 without FDR, the number of up- and down-regulated genes in response to 2 h HL increased to 108 and 23, respectively (Table 2). Many of these were also similarly regulated by exogenous treatment with Rose Bengal, or the RES acrolein and HNE (Table 2 and Appendix A).

A significant positive correlation (*r*^2^ = 0.32) was found when correlating the *log*2FC values of these treatments with HL for all 98 shared significantly affected genes (Figure 1), whereas for H_2_O_2_, the overall relationship with HL was negative (Appendix A), although only 16 genes were shared. However, it was also obvious that a proportion of genes up-regulated by HL was down-regulated by RES or Rose Bengal (Figure 1). We could not find a KEGG pathway represented by more than two genes within these 29 genes (Appendix A). A very close relationship in the regulation of significantly affected GreenCut2 genes was found when correlating *log*2FC values in response to Rose Bengal and either acrolein (*r*^2^ = 0.67; 179 genes) or HNE (*r*^2^ = 0.78; 108 genes) or H_2_O_2_ (*r*^2^ = 0.60; 46 genes), and in response to HNE and either acrolein (*r*^2^ = 0.71; 104 genes) or H_2_O_2_ (*r*^2^ = 0.62; 28 genes) (Appendix A). 

Considering only the 15 most down-regulated genes under HL, 13, 11, 5, 1 and 0 were also significantly down-regulated by Rose Bengal, 4-hydroxynonenal, acrolein, β-cyclocitral and H_2_O_2_, respectively (Figure 2). Focusing on the 15 most up-regulated genes by HL, 10, 4, 3, 0 and 0 were significantly up-regulated by acrolein, Rose Bengal, 4-hydroxynonenal, β-cyclocitral and H_2_O_2_, respectively (Figure 1). 

## 3. Discussion

### 3.1. A Role for ^1^O_2_ but Not H_2_O_2_ in Affecting the Gene Expression of GreenCut2 under HL 

To investigate how ROS contribute to the signalling of photosynthesis-specific genes during HL acclimation, we compared the differential expression of GreenCut2 genes in response to 2 h HL with 2 h treatment in response to exogenous chemicals related to ROS under LL. Previously, physiological as well as molecular responses showed that exogenous molecules (e.g., H_2_O_2_, RES and β-cyclocitral) at the concentrations investigated here were able to penetrate cells to induce a response, thus be available to act in signalling pathways [8,23,26]. For example, using fluorescent H_2_O_2_ sensors in *C. reinhardtii*, it was shown that 0.1–1.0 mM exogenous H_2_O_2_, and chloroplast-derived H_2_O_2_ under HL leaked into the cytosol [27]. Transcripts encoding the proteins involved in photosynthesis showed a general downward trend after treatment with 1 mM H_2_O_2_, but mostly insignificantly [26]. By far, the most up-regulated H_2_O_2_-responsive gene was a heat shock protein, *HSP22A* [26], which, although also highly up-regulated by Rose Bengal, acrolein and HNE, was not up-regulated in response to HL (Appendix A). This is less surprising than could be expected considering that the concentrations of H_2_O_2_ in HL-treated *C. reinhardtii* were measured at low µM [8], similar to measurements in leaves [28] and far from the 1 mM concentration used for the RNA seq. [26]. In contrast, the ^1^O_2_ gene marker, *SOUL2* [14], was significantly up-regulated by HL (Table 1) and Rose Bengal (Appendix A), supporting that ^1^O_2_ was contributing to signalling under the HL treatment. Moreover, of all GreenCut2 genes up- and down-regulated by HL, only 3% and 4%, respectively, were similarly regulated by H_2_O_2_, whereas 24% and 78%, respectively, were similarly regulated by Rose Bengal (Table 2). Worth mentioning is that the differential expression of GreenCut2 genes in response to 1 mM H_2_O_2_ was ca. 60% shared with the response to Rose Bengal (Table 1), with the *log*2FC gene expression of these treatments significantly correlating (Appendix A). Therefore, it seems that a high proportion of H_2_O_2_ signalling under such high H_2_O_2_ concentrations may pass through the same pathway(s) as ^1^O_2_. Since each ROS has distinct chemical properties, this may indicate an involvement of a common oxidative modification in response to both treatments, such as lipid oxidation and derived RES. Of all measured RES, only HNE significantly accumulated in response to 1 mM H_2_O_2_ (personal observation). However, we concluded that ^1^O_2_ has much more influence on the expression of photosynthesis-specific genes during HL acclimation than H_2_O_2_.

### 3.2. Intermediates of ^1^O_2_ Signalling Likely Include RES

The known targets of ^1^O_2_ in *C. reinhardtii* include β-carotene and PUFA, from which oxidation products could be secondary messengers in ^1^O_2_ signalling. It was shown by Roach et al. [23] that β-cyclocitral at concentrations used for the treatment (600 ppm same as used for acrolein) were also affecting the bioenergetics of the chloroplast and were thus entering the cells and available to act in chloroplast-to-nucleus signalling. However, like H_2_O_2_, the shared DEG in response to β-cyclocitral and HL was very limited (Figure 2; Table 2), thus unlikely to play a major role in HL signalling in *C. reinhardtii*. Furthermore, while β-cyclocitral down-regulated porphyrin and chlorophyll synthesis [23], HL up-regulated this pathway (Table 1). As for H_2_O_2_, the concentrations of potential ^1^O_2_-derived molecules formed under HL are also relevant for signalling. In HL-treated photoautotrophic *C. reinhardtii*, the concentration of HNE was about half that of β-cyclocitral, whereas acrolein concentrations were >5 times higher than β-cyclocitral [29]. In our study, we showed that HNE or acrolein can be mediators in ^1^O_2_ signalling of GreenCut2 genes, as revealed by the strong correlation of the *log*2FC of gene expression in response to these RES and Rose Bengal (Appendix A), but due to the higher concentration of acrolein in HL-treated *C. reinhardtii*, we expect this RES to be more involved. 

There are multiple ways ^1^O_2_ signal transduction can occur. The cytosolic phosphoprotein Sak1 [14], the zinc finger protein MBS [22], the PSII subunit P-2 (PsbP2) [30] and the ERE-containing bZIP transcription factor Sor1 [31] have all previously been shown to contribute in *C. reinhardtii*. Therefore, ^1^O_2_ signalling may pass through several of these pathways. RES-mediated ^1^O_2_ signalling was first reported in the biotic stress response of plants to pathogens [32,33]. The ^1^O_2_-induced oxidation of thylakoid membranes [11] and derived RES was shown in *C. reinhardtii* to pass through the Sor1 transcription factor [24,31]. Due to the strong positive correlation of GreenCut2 genes in response to Rose Bengal and HNE or acrolein (Appendix A), and all three treatments with HL (Figure 1), RES likely transmit the ^1^O_2_ signal from the chloroplast to the nucleus, triggering changes in gene expression under HL.

### 3.3. Up-Regulation via ^1^O_2_ of RuBisCO Activity and Down-Regulation of Glycolysis under HL 

Light intensity directly impacts photosynthetic activity and, as can be expected, up-regulated many GreenCut2 genes. Those also up-regulated by HNE, acrolein and Rose Bengal include Cre02.g120100, encoding *RBCS1* (ribulose-1,5-bisphosphate carboxylase [RuBisC] small subunit 1). Although this small subunit is not catalytic, it is essential for maximal RuBisCO activity [34]. Also upregulated by HL and Rose Bengal was Cre17.g718950, encoding *RCA2* a RuBisCO activase-like protein. Increasing CO_2_ assimilation would enhance use of HL for photosynthesis, as well as preventing ROS formation from decreasing excess light absorption. Lowered RuBisCO activity in *C. reinhardtii* mutants led to increased ROS production under HL [35], and our results support a role for ^1^O_2_-derived RES in the signalling of increasing RuBisCO activity under HL. 

Of all GreenCut2 genes down-regulated by HL, 78% were also down-regulated by RB, indicating a high level of the involvement of ^1^O_2_ signalling in lowering the transcription of unbeneficial photosynthetic processes under HL. This group includes Cre05.g232550 and Cre06.g2723, encoding *PGM4* (phosphoglycerate mutase 4) and another putative phosphoglycerate mutase, respectively, which can be involved in glycolysis by catabolising 3-phosphoglycerate (PGA), also an intermediate in the Calvin–Benson cycle. Thus, lowered PGM4 activity would increase PGA availability for inorganic carbon assimilation via RuBisCO. The involvement of ^1^O_2_ signalling in this pathway seems independent of RES (Appendix A). However, Cre10.g466500 encoding for glyoxylase I family protein was highly down-regulated by HL, as well as by HNE, acrolein and Rose Bengal. The glyoxylase pathway breaks down the products of glycolysis. Also highly down-regulated by HL, Rose Bengal and HNE was Cre09.g410700 encoding *MDH5* (chloroplastic NADP-dependent malate dehydrogenase) that converts malate to oxaloacetate. MDH5 is an oxidoreductase with NADP as a ligand and is exclusively located in chloroplasts [36]. Since MDH activity consumes NADPH [36], more NADPH would potentially be available for RuBisCO activity. Overall, the data indicate that ^1^O_2_ signalling is involved in enhancing inorganic carbon assimilation under HL.

A typical response to HL is an increase in chlorophyll *a*:*b* ratio due to less need for chlorophyll *b*-rich light-harvesting complexes (LHC). The gene Cre01.g043350 encoding *CAO1* (chlorophyllide *a* oxygenase), which plays a role in chlorophyll *b* synthesis, was down-regulated by HL, RB, HNE and acrolein, supporting a role for ^1^O_2_-derived RES in this response. 

### 3.4. Photoprotection Was Up-Regulated by Acrolein Without HL 

The dissipation of excess light energy to heat via NPQ is a universal strategy of photosynthetic organisms activated by HL [9]. In *C. reinahardtii*, NPQ via LHC-stress-related (LHCSR) proteins protects from ^1^O_2_ formation and photoinhibition [23,29]. Of the 15 GreenCut2 genes most up-regulated by HL, 10 were also significantly up-regulated by acrolein, despite acrolein treatments being made under LL (Table 2). Included in this group are Cre01.g016750 encoding a PSBS protein, Cre01.g016600 encoding *PSBS1* (the two most up-regulated GreenCut2 genes), Cre09.g394325 encoding *ELI3* (Early light-inducible protein) and Cre02.g109950 encoding *HLIP* (single-helix LHC light protein). In *A. thaliana*, *ELIP2* (*ELI3* homologue in *C. reinhardtii*) was shown to be involved in various stress responses, such as cold and UV-B [37], while *PSBS1* is required for the activation of NPQ, possibly by promoting the conformational changes needed for the activation of LHCSR-dependent quenching in the antenna of PSII [38,39]. Although not a GreenCut2 gene, *LHCSR1* is also strongly up-regulated by HL and acrolein [24]. Overall, acrolein seems to have an important function in the acclimation process against HL by up-regulating photoprotection. 

## 4. Materials and Methods

### Material Source and Origin of Data

*Chlamydomonas reinhardtii* wild-type (WT) strain 4A mt^+^ (CC-4051) was used for all analyses. For treatments, axenic cultures were established in TAP (TRIS-acetate-phosphate) media under LL at 50 μmol quanta m^−2^ s^−1^ (sub-saturating), and once in exponential phase, were transferred to photoautotrophic THP media or placed on solid agar TAP media, depending on the treatment. For volatile treatments (acrolein, β-cyclocitral), homogenous algal ‘lawns’ were initiated on agar by distributing 0.75 mL of liquid TAP culture evenly across 11 cm Petri dishes half-filled with TAP +1.5% agar and left for 0.5 h in a laminar flow bench to evaporate the liquid media in sterile air. The lid was then closed but not sealed, and the cells were cultivated for 4 days at 20 °C under LL before treatment with volatile acrolein or β-cyclocitral (both Sigma-Aldrich, St. Louis, MI, USA). For more details, see [23,24]. For treatments of liquid cultures (HL, Rose Bengal, HNE), TAP cultures were transferred to a photoautotrophic THP medium, whereby the media was pH-adjusted to 7.0 with HCl rather than with acetic acid. The cells were pelleted at 1000× *g* for 1 min and TAP media was replaced with THP. Liquid THP cultures were bubbled with sterile air, achieved with a 0.2 μM air-filter (Minisart NML Plus cellulose acetate filters; Sartorius, Göttingen, Germany) and cultivated for at least 24 h under LL with a culture rotation at 75 rpm before treatment. The concentration of HNE (>98% pure, Cayman Chemical Co. Ann Arbor, MI, USA) in the culture was 37.5 µM with 14.6 µL of 10 mg/mL ethanol stock solution added to 25 mL culture under LL, which was calculated to be the same concentration as for the volatile acrolein treatment [24]. Cells were treated with 1 µM Rose Bengal dissolved in H_2_O (95% pure disodium salt, Sigma-Aldrich) under LL, which provides a tolerable dose of ^1^O_2_ [23]. For HL treatment, the light intensity was increased to 750 μmol quanta m^−2^ s^−1^ (*ca.* double the saturating light intensity for *C. reinhardtii* [7]). Liquid cultures were at a density of 15 µg chlorophyll ml^−1^ for all samples, and during all treatments, they were rotated, but no longer bubbled with air. For the analysis of differential gene expression, comparisons were either made to the respective non-treated liquid or agar-grown cultures from LL. Three separate cultures were used as replicates for controls and treatments. 

Total RNA was extracted with the RNeasy Plant Mini Kit (Qiagen, Hilden, Germany) and additional on-column DNAse treatment (RNAse-free DNAse set, Qiagen) according to the manufacturer’s instructions. Briefly, 15 mg of agar culture carefully scraped from the surface or 25 mg of pelleted liquid culture (after centrifugation for 1 min at 1000× *g*) were frozen with three 3 mm RNAse free glass beads in a 2 mL Eppendorf tube and stored at –80 °C. They were then shaken in pre-ice-cooled adaptors for 2 min at 30 Hz (TissueLyzer II, Qiagen) and 450 µL of an RLC buffer with β-mercaptoethanol was added immediately. After extraction, the samples were stored at –80 °C before the poly A enrichment of mRNA. The RNA seq. was performed by the NGS Core Facility of the Vienna Biocenter, Vienna, Austria, with Illumina’s HiSeq2500 instrument using single-end sequencing with 50 bp read length. Raw reads were aligned against the *C. reinhardtii* reference genome (JGI v5.5 release) with STAR version 2.5.1b using a 2-pass alignment mode. Three biological replicates were analysed for each treatment and a significant FC was considered at *p* < 0.05 with or without FDR, as calculated with the Limma package [40].

The Algal Functional Annotation Tool was used (http://pathways.mcdb.ucla.edu/algal/index.html, accessed on 4 June 2024) for exploring the KEGG pathways of shared DEGs. 

## 5. Conclusions

Previously, we have shown that ^1^O_2_ formation during HL peroxidises PUFA in thylakoid membranes, leading to the release of RES. Here, evidence is provided that RES, such as acrolein and HNE, can be the secondary messengers of ^1^O_2_ signalling, up-regulating photo-protection and possibly carbon assimilation, while down-regulating specific primary metabolic pathways towards HL acclimation. In contrast, we found that H_2_O_2_ and β-cyclocitral most likely do not contribute. 

## Figures and Tables

**Figure 1 ijms-25-08458-f001:**
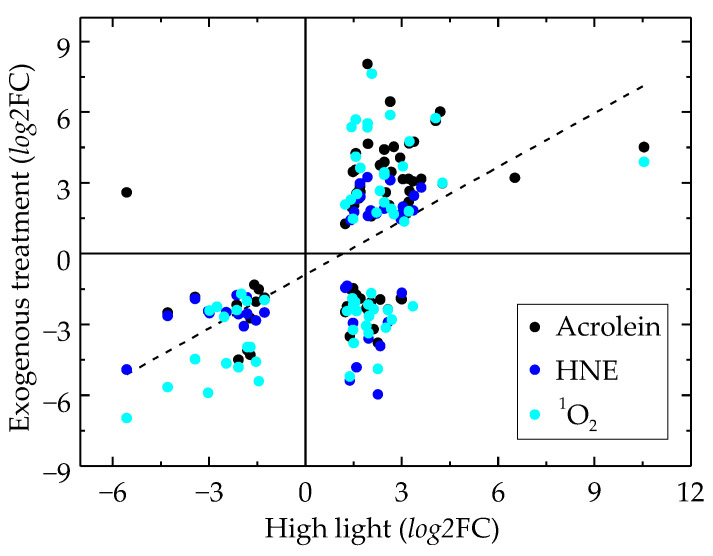
Correlation of the expression levels of GreenCut2 genes significantly affected by high light (HL) and singlet oxygen (^1^O_2_)-related treatments under low light. Data are separated along the *x*-axis by a *log*2-fold change (*log*2FC) in response to HL and along the *y*-axis by *log*2FC in response to exogenous treatments (acrolein; black, HNE; dark blue, Rose Bengal (^1^O_2_); turquoise). The line of best fit is of all data.

**Figure 2 ijms-25-08458-f002:**
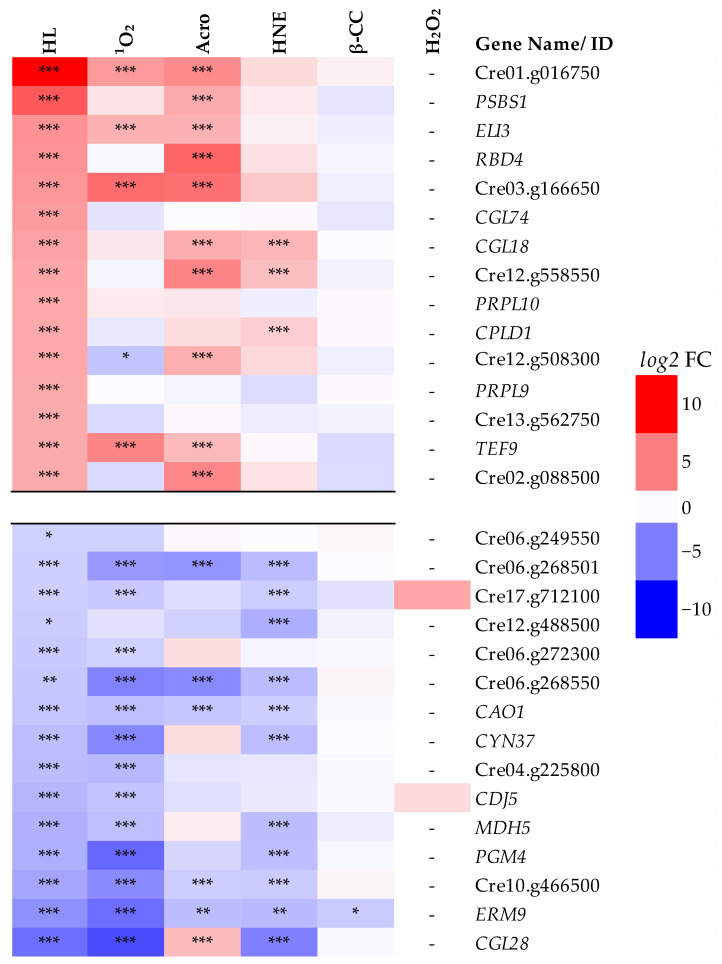
Fold-change (FC) of the 15 most up- and down-regulated GreenCut2 genes in response to high light (HL) and exogenous treatments under low light. Red and blue indicate up- and down-regulation, respectively on a *log*2FC scale shown on the right. When known, the gene names are given, otherwise the gene ID is denoted. HL: high light, ^1^O_2_: Rose Bengal, Acro: acrolein, HNE: 4-hydoxynonenal, β-CC: β-cyclocitral, H_2_O_2_: hydrogen peroxide. Significance: *p* < 0.05, *p* < 0.05 + FDR and *p* < 0.01 + FDR are denoted by *, ** and ***, respectively. Dash (-): insignificant [26].

**Table 1 ijms-25-08458-t001:** GreenCut2 genes significantly affected by 2 h HL. Significance was considered by *p* < 0.05 with FDR and >2-fold change (FC) relative to LL conditions, i.e., >1 or <−1 after *log*2 transformation. Red and blue shading of FC values denotes up- and down-regulation, respectively. When known, the gene name, protein description and putative or likely function are provided.

JGI v5.5 ID	FC (*log*2)	Gene	Description	Function
Cre01.g016750	10.5	*-*	Photosystem II 22 kDa protein (psbS) (1 of 2)	NPQ (likely)
Cre01.g016600	6.5	*PSBS1*	Chloroplast PSII-associated 22 kDa protein	NPQ
Cre09.g394325	4.3	*ELI3*	Early-light-inducible protein	High light stress response
Cre12.g510400	4.2	*RBD4*	Putative rubredoxin-like protein	
Cre03.g166650	4.1	*-*	DEAD/DEAH-box DNA/RNA helicase	
Cre12.g554103	3.8	*CGL74*	-	
Cre03.g145567	3.6	*CGL18*	-	
Cre12.g558550	3.4	*-*	Hydrolase, alpha/beta fold family protein	
Cre06.g272850	3.4	*PRPL10*	Plastid ribosomal protein L10	Plastid protein synthesis
Cre03.g145587	3.4	*CPLD1*	Putative thiol-disulphide oxidoreductase DCC	PSII assembly (likely)
Cre12.g508300	3.3	*-*	Protein of unknown function (DUF493)	
Cre12.g556050	3.3	*PRPL9*	Plastid ribosomal protein L9	Plastid protein synthesis
Cre13.g562750	3.2	*-*	Domain of unknown function (DUF4336)	
Cre12.g519300	3.2	*TEF9*	-	
Cre02.g088500	3.2		Conserved expressed protein	
Cre06.g265800	3.2	*PRPL28*	Plastid ribosomal protein L28	Plastid protein synthesis
Cre13.g580300	3.2	*-*	ABC transporter family protein	
Cre02.g109950	3.1	*HLIP*	Single-helix LHC light protein	High light stress response
Cre13.g579550	3.1	*CGL27*	-	
Cre12.g509650	3.1	*PDS1*	Phytoene desaturase	Carotenoid biosynthesis
Cre12.g498550	3.0	*CHLM*	Magnesium protoporphyrin O-methyltransferase	Chlorophyll biosynthesis
Cre03.g199535	3.0	*-*	Low-molecular-mass early-light-induced protein	Chlorophyll biosynthesis
Cre01.g049000	3.0	*-*	Pterin dehydratase	
Cre05.g242000	3.0	*CHLD*	Magnesium chelatase subunit D	Chlorophyll biosynthesis
Cre12.g494750	3.0	*PRPS20*	Plastid ribosomal protein S20	Plastid protein synthesis
Cre01.g004450	2.9	*CPLD42*	Membrane protein	
Cre02.g145000	2.9	*-*	K02834 ribosome-binding factor A (rbfA)	
Cre09.g411200	2.8	*TEF5*	Rieske [2Fe-2S] domain-containing protein	Stability of PSII–LHCII
Cre17.g721700	2.7	*CPLD44*	Thylakoid luminal protein	
Cre07.g334600	2.7	*CGL20*	-	
Cre02.g120100	2.6	*RBCS1*	RuBisCO small subunit 1	CO_2_ assimilation
Cre12.g541777	2.6	*-*	Ribosomal n-lysine methyltransferase 3	CO_2_ assimilation
Cre05.g246800	2.6	*GUN4*	Tetrapyrrole-binding protein	Chlorophyll biosynthesis
Cre03.g150350	2.6	*-*	KOG1803—DNA helicase	
Cre03.g195200	2.5	*-*	Haloalkane dehalogenase-like hydrolase	
Cre18.g748397	2.5	*-*	*-*	
Cre12.g490500	2.5	*CGL78*	-	
Cre10.g440450	2.5	*PSB28*	Photosystem II subunit 28	PSII biogenesis (likely)
Cre12.g510850	2.5	*CGL73*	-	
Cre03.g175200	2.3	*TOC75*	Translocon; outer envelope membrane of chloroplasts	
Cre13.g562850	2.3	*THF1*	Thylakoid formation protein	
Cre10.g466850	2.3	*FKB18*	Peptidyl-prolyl cis-trans isomerase, FKBP-type	
Cre08.g372000	2.2	*CGLD11*	-	ATP synthesis
Cre07.g328200	2.2	*PSBP6*	Lumen-targeted protein	
Cre03.g151200	2.2	*CGLD16*	-	
Cre03.g176350	2.2	*PLP5*	Plastid-lipid-associated protein	Acyl-lipid metabolism (likely)
Cre14.g629650	2.1	*NIK1*	Nickel transporter	
Cre06.g252200	2.1	*TOC34*	Translocon; outer envelope membrane of chloroplasts	
Cre09.g398700	2.1	*CFA2*	Cyclopropane-fatty-acyl-phospholipid synthase	
Cre03.g165000	2.1	*LPA1*	Translation elongation factor EFG/EF2	PSII assembly
Cre12.g537850	2.1	*CCB2*	Protein required for cyt b6 assembly	Cytochrome b6 assembly
Cre02.g082300	2.0	*-*	Surfeit locus protein 6	
Cre01.g015950	2.0	*CPL11*	Translation factor	Plastid protein synthesis
Cre03.g145207	2.0	*CPLD33*	-	
Cre06.g251150	2.0	*OHP1*	Low-CO_2_ and stress-induced one-helix protein	PSII assembly
Cre12.g530300	2.0	*-*	Peptidyl-prolyl cis-trans isomerase, FKBP-type	
Cre13.g578650	2.0	*HCF173*	Similar to complex I intermediate-associated protein 30	PSII assembly (likely)
Cre16.g670950	2.0	*CYC4*	Chloroplast cytochrome c	Redox
Cre16.g679300	2.0	*-*	*-*	
Cre01.g042800	1.9	*DVR1*	3,8-divinyl protochlorophyllide a 8-vinyl reductase	Chlorophyll biosynthesis
Cre13.g566850	1.9	*SOUL2*	SOUL heme-binding protein	Heme binding
Cre13.g570350	1.9	*AKC4*	ABC1-like kinase	
Cre10.g438550	1.9	*TAT1*	TatA-like sec-independent protein translocator	Protein transport
Cre06.g261500	1.9	*-*	Thioredoxin family protein	Redox
Cre16.g673550	1.9	*-*	S-methyl-5-thio-D-ribose-1-phosphate isomerase	Methionine metabolism
Cre01.g002250	1.9	*-*	Acyl-CoA n-Acyltransferase domain-containing	
Cre06.g294750	1.9	*CHLG*	Chlorophyll synthetase	Chlorophyll biosynthesis
Cre03.g157800	1.8	*-*	Thioredoxin-like protein	Redox
Cre07.g315150	1.7	*RBD1*	Rubredoxin	PSII assembly
Cre07.g329000	1.7	*CPLD47*	Predicted membrane protein	PSII assembly
Cre12.g500650	1.7	*RNB2*	3-5 exoribonuclease II	RNA processing
Cre16.g666050	1.7	*-*	Saccharopine dehydrogenase	Cyt. *b*_6_*f* assembly
Cre01.g000850	1.6	*CPLD38*	-	Stability of Cyt. *b_6_f*
Cre12.g498700	1.6	*CPLD13*	-	
Cre09.g416200	1.6	*MBB1*	PsbB mRNA maturation factor, chloroplastic	PSII assembly
Cre06.g278236	1.6	*-*	Ubiquinone/menaquinone methyltransferase	
Cre01.g021600	1.6	*-*	RNA helicase//subfamily not named	
Cre06.g269300	1.6	*-*	PF07103 —protein of unknown function (DUF1365)	
Cre17.g720050	1.6	*FHL2*	FtsH-like membrane ATPase/metalloprotease	
Cre02.g095097	1.6	*-*	Peptidyl-prolyl cis-trans isomerase, FKBP-type	
Cre16.g661150	1.5	*CGL5*	-	Carotenoid modification
Cre03.g182150	1.5	*TEF8*	-	PSII assembly (likely)
Cre06.g296250	1.5	*-*	Lysyl-tRNA synthetase	
Cre01.g052050	1.5	*-*	Ubiquinol-cytochrome C chaperone	Cyt. *b* assembly
Cre03.g184550	1.5	*CPLD28*	-	PSII assembly (likely)
Cre16.g665250	1.4	*APE1*	Thykaloid-associated protein	Acclimation to variable light
Cre02.g114750	−1.4	*-*	MAP kinase-activated protein kinase 5	Protein phosphorylation
Cre05.g248000	−1.5	*CGL29*	-	
Cre12.g540500	−1.6	*-*	Peroxisomal membrane protein pmp27	
Cre12.g543000	−1.7	*-*	-	
Cre08.g379350	−1.7	*TPT1*	Triose phosphate transporter	Sugar transporter
Cre06.g268501	−1.8	*-*	2-5 RNA ligase superfamily	
Cre17.g712100	−1.8	*MDAR1*	Pyridine nucleotide–disulphide oxidoreductase	Redox
Cre06.g272300	−2.0	*-*	Phosphoglycerate mutase family protein	Glycolysis (potential)
Cre06.g268550	−2.1	*-*	Glucomannan 4-beta-mannosyltransferase	
Cre01.g043350	−2.1	*CAO1*	Chlorophyllide a oxygenase	Chlorophyll *b* synthesis
Cre06.g303300	−2.5	*CYN37*	Putative peptidyl-prolyl cis-trans isomerase	
Cre04.g225800	−2.5	*-*	Ankyrin repeat protein	
Cre07.g320350	−2.8	*CDJ5*	Chloroplast DnaJ-like protein	
Cre09.g410700	−3.0	*MDH5*	NADP-dependent malate dehydrogenase	Organic acid metabolism
Cre05.g232550	−3.0	*PGM4*	Phosphoglycerate mutase	Glycolysis
Cre10.g466500	−3.4	*-*	Glyoxylase I family protein	Glycolysis (potential)
Cre10.g460150	−4.3	*ERM9*	ERD4-related membrane protein	
Cre10.g439700	−5.6	*CGL28*	RNA-binding protein	

**Table 2 ijms-25-08458-t002:** Number of differentially expressed GreenCut2 genes and the % shared amongst treatments. Numbers below the arrows indicate total up-(↑) and down-(↓)regulated GreenCut2 genes in response to treatments listed above (*p* < 0.05 without FDR). Numbers in the table indicate % of the total ↑ or ↓ DEG listed on top that were also ↑ or ↓ regulated, respectively, by the treatments listed vertical–left. HL: high light, ^1^O_2_: Rose Bengal, HNE: 4-hydoxynonenal, β-CC: β-cyclocitral, H_2_O_2_: hydrogen peroxide.

Total DEG	HL		^1^O_2_		Acrolein		HNE		β-CC		H_2_O_2_	
	↑ 108	↓ 23	↑ 140	↓ 180	↑ 138	↓ 144	↑ 53	↓ 114	↑ 2	↓ 14	↑ 32	↓ 31
**HL**	-	-	19	10	28	8	34	12	0	7	9	3
**^1^O_2_**	24	78	-	-	59	57	55	71	50	43	59	61
**Acrolein**	36	48	58	46	-	-	72	59	50	50	56	52
**HNE**	16	61	21	45	28	47	-	-	0	43	25	52
**β-CC**	0	4	1	3	1	5	0	5	-	-	3	19
**H_2_O_2_**	3	4	14	11	13	11	15	14	50	43	-	-

## Data Availability

RNA seq. data are available at the Sequencing Read Archive of NCBI project PRJNA1123657 https://www.ncbi.nlm.nih.gov/sra/PRJNA1123657.

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
