# Peer review of "Singlet-Oxygen-Mediated Regulation of Photosynthesis-Specific Genes: A Role for Reactive Electrophiles in Signal Transduction"

_ijms, 2024, doi:10.3390/ijms25158458_

Round 1

Reviewer 1 Report

Comments and Suggestions for Authors

The authors performed an RNA-seq study to investigate the role of reactive oxygen species in alga Chlamydomonas reinhardtii. I can see the value of this study, but a lot of information is missing, and the manuscript needs to be improved for this work to be published. 

Major comments:

1.     The experiment design is not clear/ well-described

2.     Material and methods are missing essential information for the experiments

3.     The authors did a good amount of work on RNA-seq sequencing for multiple samples, but the analyses were not sufficient

Specific comments:

Introduction:

1.     Provide more description of the GreenCut2 gene database, specifically that it includes algal genes. It is not a database that everyone is familiar with.

2.     Provide a justification for using external sources of ROS for this study (the authors mentioned it in the discussion, but this should also be mentioned here to inform the readers).

Results/ discussion:

3.     Why did the authors use p <= 0.05 instead of p<0.05? This is also not consistent between results and methods.

4.     Line86- no FDR correction, but Line 88- with FDR. This needs to be consistent, and the authors need to justify why there is no FDR correction. The authors also did not provide any cut-off values for fold-change. This needs to be specified.

5.     Table 1: Bars were used for TPM values, but there were no scales (e.g. x-axis if this was a bar graph). It would be more straightforward for the readers to understand if Log2(Fold-change) was converted back to fold-change values. Essential genes described in Lines 95-100 and elsewhere could be highlighted in the table.

6.     Line 107- wrong spell

7.     Table 2 is very hard to read and understand. I would suggest the authors consider other ways to show the data. The spelling of the horizontal and vertical names need to be consistent (e.g. Acro vs. acro, bCC vs. BCC…)

8.     Figure 1. Is the scale actually fold-change or log2 fold-change? It is not clear what the stars (*) mean and what the difference is between the different numbers of stars. It would also be helpful to provide a description of the genes as in Table 1 instead of just abbreviated gene names. 

9.     More analyses should be included. It would be good to include things like pathway analyses/ gene ontology enrichment analyses

Materials and method 

10.  This section did not include sufficient information for the experiments

11.  A section should be included to describe the condition for stock cultures (culture source, temperature, nutrients…). Was the culture xenic or axenic?

12.  Manufacturer’s information should be provided for all compounds for the treatment

13.  It is not clear which light level is the low light, and how long the cultures were treated under the high light condition. If 50 μmol quanta m−2 s−1 was the “low light” condition, the authors need to be clear. And is this a regular light condition or really a low light for their stock cultures? In the manuscript elsewhere, it mentioned the culture was treated under high light for 2 hours. This should be provided in the method. It was also not clear if the chemical treatment was under low or high light conditions in the method section.

14.  Line 244: culture was grown with rotation indicates the cultures were in liquid medium, but then Line 246 mentioned agar-grown cultures. If the culture condition was changed, it should be clearly described.

15.  Line 244: using mg chlorophyll ml–1 to describe the culture is odd. How was it measured? Was this measured by in vivo fluorescence? Did the authors mean at a chlorophyll level of x mg/ mL? 

16.  Were the concentrations for compounds provided final concentrations in the cultures? What were they dissolved in? Was the same amount of solvent added to the control culture?

17.  Growth data were missing for the control and treatments

18.  It is not clear what the control was and how many replicates there were for each of the controls and treatments.

19.   Line 247: Why did the authors use Rose Bengal at much lower concentrations compared to other compounds?

20.  Line 249- 256: Lots of very important information is missing for the RNA-seq data. How were the samples collected? How was RNA extracted?  What was the quality control step? Though the sequencing was performed in another facility, general steps should be included (e.g. protocol/ methods used…). How were the sequences annotated? Were they only annotated against GreenCut2 genes or other databases were used? Citations for methods and software should be provided.

Author Response

Please see the attched file

Reviewer 2 Report

Comments and Suggestions for Authors

The manuscript, “Singlet oxygen mediated regulation of photosynthesis-specific 2 genes: A role for reactive electrophiles in signal transduction,” by Pancheri et al., is mostly well-written. The topic is relevant to elucidating the high-light acclimation response of Chlamydomonas cells. A couple of shortcomings are mentioned below.

1.     The manuscript fails to explain or compare the gene expression differences observed between the RES; acrolein and 4-hydroxynonenal. The authors should discuss this.

2.     The Methods section does not mention the methodology used for RNA isolation.
